# Inhibition of Indoxyl Sulfate-Induced Reactive Oxygen Species-Related Ferroptosis Alleviates Renal Cell Injury In Vitro and Chronic Kidney Disease Progression In Vivo

**DOI:** 10.3390/antiox12111931

**Published:** 2023-10-30

**Authors:** Li-Ting Tsai, Te-I Weng, Ting-Yu Chang, Kuo-Cheng Lan, Chih-Kang Chiang, Shing-Hwa Liu

**Affiliations:** 1Institute of Toxicology, College of Medicine, National Taiwan University, Taipei 100, Taiwan; f10447003@ntu.edu.tw (L.-T.T.); d03447003@ntu.edu.tw (T.-Y.C.); ckchiang@ntu.edu.tw (C.-K.C.); 2Department of Forensic Medicine, College of Medicine, National Taiwan University, Taipei 100, Taiwan; wengtei2@ntu.edu.tw; 3Department of Emergency Medicine, Tri-Service General Hospital, National Defense Medical Center, Taipei 114, Taiwan; 4Departments of Integrated Diagnostics & Therapeutics and Internal Medicine, College of Medicine and Hospital, National Taiwan University, Taipei 100, Taiwan; 5Department of Medical Research, China Medical University Hospital, China Medical University, Taichung 404, Taiwan; 6Department of Pediatrics, College of Medicine, National Taiwan University & Hospital, Taipei 100, Taiwan

**Keywords:** indoxyl sulfate, ferroptosis, ROS, fibrosis, chronic kidney disease

## Abstract

The accumulation of the uremic toxin indoxyl sulfate (IS) is a key pathological feature of chronic kidney disease (CKD). The effect of IS on ferroptosis and the role of IS-related ferroptosis in CKD are not well understood. We used a renal tubular cell model and an adenine-induced CKD mouse model to explore whether IS induces ferroptosis and injury and affects iron metabolism in the renal cells and the kidneys. Our results showed that exposure to IS induced several characteristics for ferroptosis, including iron accumulation, an impaired antioxidant system, elevated reactive oxygen species (ROS) levels, and lipid peroxidation. Exposure to IS triggered intracellular iron accumulation by upregulating transferrin and transferrin receptors, which are involved in cellular iron uptake. We also observed increased levels of the iron storage protein ferritin. The effects of IS-induced ROS generation, lipid peroxidation, ferroptosis, senescence, ER stress, and injury/fibrosis were effectively alleviated by treatments with an iron chelator deferoxamine (DFO) in vitro and the adsorbent charcoal AST-120 (scavenging the IS precursor) in vivo. Our findings suggest that IS triggers intracellular iron accumulation and ROS generation, leading to the induction of ferroptosis, senescence, ER stress, and injury/fibrosis in CKD kidneys. AST-120 administration may serve as a potential therapeutic strategy.

## 1. Introduction

Chronic kidney disease (CKD) is a major public health problem worldwide. The worldwide prevalence of CKD is estimated to be approximately 13.4% [1]. Uremic toxins are substances that accumulate in the body due to impaired kidney function. Among these toxins, indoxyl sulfate (IS), derived from tryptophan metabolism accumulation [2], has been associated with various pathological features of CKD, including oxidative stress, inflammation, endothelial dysfunction, and vascular calcification [3,4,5,6]. Furthermore, IS has been linked to the development of cardiovascular disease, a major cause of mortality in CKD patients [7]. Previous studies have shown that IS accelerates senescence and induces ER stress in renal tubular cells through the production of reactive oxygen species (ROS) [8,9], which has been demonstrated to play important roles in multiple diseases. 

The adenine-induced CKD model has been widely recognized as a valuable tool for studying renal injury, which is characterized by loss of renal function, such as proteinuria, increased blood urea nitrogen (BUN), uric acid, creatinine in serum, and renal tubular injury, including an increasing number of dilated tubules or cellular debris in the tubular lumen and inducing fibrosis [10,11,12]. This model offers a reliable way to study CKD pathophysiology and develop new treatments. AST-120 is an orally administered charcoal-based adsorbent that works by absorbing indole produced in the gastrointestinal tract, thereby reducing IS concentrations in serum and urine [13]. AST-120 has demonstrated clinical advantages in preventing CKD progression and delaying the need for dialysis when combined with standard therapy [14]. 

Ferroptosis is a unique form of cell death driven by iron-dependent phospholipid peroxidation [15]. The characteristics of ferroptosis include the accumulation of intracellular iron levels, impaired production of antioxidants, such as GPX4 and cystine/glutamate antiporter SLC7A11 (xCT), and the occurrence of lipid peroxidation [16]. Recent studies have established a close association between ferroptosis and the pathophysiology of various diseases, including tumors, nervous system disorders, ischemia–reperfusion injury, kidney damage, and blood disorders [16,17]. Consequently, there is growing interest in how to intervene in the development and occurrence of these conditions by regulating cell ferroptosis. Iron is an essential element in the construction of various important macromolecules, such as those involved in metabolism, energy production, DNA synthesis, and respiration [18]. However, excessive labile iron can be harmful to cells or organisms, as it can lead to the formation of damaging free radicals via oxidation-reduction reactions such as the Fenton reaction [19,20]. Previous studies have demonstrated elevated renal iron accumulation and urinary iron excretion in CKD patients and animal models of renal disease [21]. Nevertheless, the effects of IS on ferroptosis and the metabolic mechanisms of iron in the kidneys are not fully understood.

This study was designed to examine whether IS induces ROS-related ferroptosis and its impact on iron metabolism in renal cells. We studied the role of ferroptosis in the IS-induced renal injury using a renal tubular cell model and an adenine-induced CKD mouse model. The research not only illustrated the mechanisms through which IS induced ferroptosis but provided a promising therapeutic option for CKD patients. 

## 2. Materials and Methods

### 2.1. Chemicals and Reagents

Chemicals for indoxyl sulfate (IS), N-acetylcysteine (NAC), and adenine were acquired from Sigma-Aldrich (St. Louis, MO, USA). AST-120 (scavenging the IS precursor) (KREMEZIN) was obtained from Conmed Pharmaceutical & Bio-Medical Corporation (Taipei, Taiwan). Iron chelator deferoxamine (DFO) was purchased from Cayman Chemical Company (Ann Arbor, MI, USA). 

### 2.2. Cell Culture and Treatments

A normal rat kidney epithelial cell line (NRK-52E) obtained from the American Type Culture Collection (ATCC; Manassas, VA, USA) was used. The cells were cultivated as adherent cells in DMEM medium supplemented with 5% FBS, 1% penicillin–streptomycin–amphotericin B, and NaHCO3 (1.5 g/L). For each experiment using cultured NRK-52E cells treated with or without IS (0.1–1 mM), the plating density was 1 × 10^4^ cells/well for 96-well plates and 3 × 10^5^ cells/well for 6-well plates. In some experiments, NRK-52E cells were pretreated with N-acetylcysteine (NAC, dissolved in PBS, 1 mM) or DFO (dissolved in PBS, 50 µM) for 1 h, followed by treatment with IS for 24 h. These same plating conditions were used in all experiments unless mentioned otherwise.

### 2.3. Animal Model and Treatment Strategies

Male 6-week-old C57BL/6 mice obtained from the Laboratory Animal Center of the National Taiwan University College of Medicine were used. The animal experiments were performed in accordance with the Guide of the Association for Assessment and Accreditation of Laboratory Animal Care International (AAALAC). The Institutional Animal Care and Use Committee at the National Taiwan University College of Medicine (Taipei, Taiwan) approved the animal protocol and procedures. The experimental mice were randomly grouped as follows: (1) control; (2) adenine (50 mg/kg body weight); and (3) adenine + AST-120 (4 mg/g body weight/day). Adenine in 0.5% CMC was administered to mice (50 mg/kg body weight) via oral gavage daily for 28 days to induce CKD [22]; after 14 days of adenine administration, the mice were then given AST-120 at a dose of 4 mg/g body weight/day via oral gavage for 14 days. Six mice per group were used. All of the mice were euthanized on day 28, and the serum and kidney samples were collected for further analysis.

### 2.4. Levels of Serum Creatinine, Blood Urea Nitrogen (BUN), and Indoxyl Sulfate

The levels of serum creatinine and BUN in mice were tested using a clinical chemistry analyzer (Roche, Rotkreuz, Switzerland) as previously described [22]. The serum indoxyl sulfate levels in mice were determined by the methods of ultra-high-performance liquid chromatography (UHPLC; Agilent 1290 infinity II, Santa Clara, CA, USA) and tandem mass spectrometry (LC-MS/MS; Sciex QTRAP 6500, Framingham, MA, USA) as previously described [22].

### 2.5. Histological Analysis

Fresh kidney samples were preserved in 10% formalin, followed by embedding in paraffin and sectioning into 4 μm thick slices. Renal histological injury was assessed using H&E-stained sections. Masson’s trichrome-stained sections were used to evaluate renal fibrosis, and PAS stain was used to detect the presence of complex carbohydrates, glycogen, mucin, and other substances in tissues. The positive staining was measured by ImageJ at 400× magnification.

### 2.6. Immunohistochemistry (IHC) Staining

An IHC kit (cat. no. ab64264, Abcam, Cambridge, UK) was used to conduct the IHC staining. Kidney tissue sections were deparaffinized and gradually rehydrated using a series of ethanol solutions. To facilitate antigen retrieval, a protease reagent was used to treat the slides and, subsequently, a protein block reagent was used to block the slides for 10 min. Finally, the sections were incubated with the appropriate antibodies: p53 (1:50; cat. no. sc-126, Santa Cruz Biotechnology, Dallas, TX, USA), p16 (1:100; cat. no. sc-1661, Santa Cruz), p21 (1:100; cat. no. ab188224, Abcam), beta-gal (1:100; cat. no. 15518-1-AP, Proteintech, Rosemont, IL, USA), GPX4 (1:100; cat. no. ab125066, Abcam), HO-1 (1:100; cat. no. GTX101147, GeneTex, Alton Pkwy Irvine, CA, USA), 4-HNE (1:100; cat. no. GTX101147, GeneTex), and FTH (1:100; cat. no. GTX101147, GeneTex) at 4 °C overnight. The slides were treated with the HRP-conjugated secondary antibodies and then incubated with DAB substrate. The sections were counterstained with hematoxylin and mounted using a mounting medium. Subsequently, the slides were analyzed using ImageJ software (version 1.48) with the IHC profiler plugin. The IHC measurement index was calculated utilizing the average gray value of positive cells (staining intensity) and the percentage of positive areas (staining area) as parameters. Based on this index, the scoring system assigned four categories: high positive (3+), positive (2+), low positive (1+), and negative (0).

### 2.7. Iron Stain

Formalin-fixed, paraffin-embedded kidney tissue sections were stained with an iron stain kit (Prussian Blue Stain; cat. no. ab150674, Abcam) to detect cellular iron accumulation. Briefly, potassium ferrocyanide solution and hydrochloric acid solution were combined in equal proportions. Deparaffinized and rehydrated sections were immersed in iron stain solution for 30 min and then counterstained with nuclear fast red for 5 min. The positive staining was measured by ImageJ software (version 1.48).

### 2.8. Immunoblotting and Antibodies

The cell lysis was prepared using the RIPA lysis buffer, which had a protease inhibitor cocktail (cat. no. 78430, ThermoFisher Scientific, Waltham, MA, USA) added. Kidney tissues were homogenized in RIPA buffer containing the protease inhibitor. After that, cell extracts and tissue homogenates were subjected to centrifugation at 13,000 rpm for 30 min at 4 °C to eliminate cell debris. Then, the supernatants containing the protein extracts were collected. The concentrations of proteins were determined using a BCA Protein Assay kit (PierceTM, ThermoFisher Scientific). The samples were loaded and separated via SDS-polyacrylamide gel electrophoresis, transferred onto a PVDF membrane (Merck Millipore, Billerica, MA, USA), and subsequently blocked using 5% BSA. Primary antibodies were applied, followed by secondary antibodies. An enhanced ECL system (Bio-Rad, Hercules, CA, USA) was used to detect the immunoreactive proteins. Antibodies for xCT (ab175186), GPX4 (ab125066), p21 (ab109199), PTGS2 (ab15191), ATF6 (ab2031119), ATF4 (ab11815), and GAPDH (ab181602) were purchased from Abcam; eIF2⍺ (5324) and CHOP (2895) were obtained from Cell Signaling Technology (Danvers, MA, USA). HO-1 (GTX101174), Tf (GTX101035), TfR (GTX102596), FTH (ferritin heavy chain; GTX101733), NCOA4 (GTX32739), FPN (GTX54821), and DMT1 (GTX64686) were provided by GeneTex. Nrf2 (sc-722), Keap-1 (sc-115246), p16 (sc-1661), p53 (sc-126), GRP78 (sc-13968), and p-eIF2α (sc-101670) were obtained from Santa Cruz. 

### 2.9. MTT Assay

A 3-(4,5-dimethylthiazol-2-yl)-2,5-diphenyltetrazolium bromide (MTT; Sigma-Aldrich) assay was used to determine the cell viability. NRK-52E cells were seeded at a density of 2 × 10^4^ cells/well in 96-well plates and allowed to adhere and recover overnight. The cells were then treated with IS (0.1–1 mM) for 24 h after changing to fresh media. Following the incubation period, the medium was removed, and 10% MTT solution was added to each well. Viable cells reduced MTT to formazan, and then DMSO (100 μL) was used to dissolve the insoluble formazan crystals. A microplate reader (model 550, Bio-Rad) was used to detect the absorbance at 570 nm.

### 2.10. DCFDA Cellular ROS Detection Assay

The content of ROS was determined via a DCFDA/H2DCFDA-Cellular ROS Assay Kit (cat. no. ab113851, Abcam). Briefly, the cells were seeded in 96-well plates overnight. Subsequently, the cells were transferred to fresh media containing IS and incubated for 6 h. Then, the 2′,7′-dichlorofluorescin diacetate (DCFDA; 25 µM) was added to the cells for 45 min. After washing with buffer, fluorescence was detected via a fluorescence microplate reader at Ex/Em of 485 nm/535 nm.

### 2.11. Senescence-Associated β-Galactosidase (SA-β-gal) Staining

The SA-β-gal staining was determined using a Senescence Detection Kit (BioVision, Cambridge, U.K.). NRK-52E cells (3 × 10^5^ cells/well) cultured in 6-well plates were treated with IS for 24 h, and then washed with phosphate-buffered saline (PBS; 1 mL/well) and fixed with fixative solution (1 mL/well) for 10 min at room temperature. The fixed cells were then stained with a 1 mL/well mixture of staining solution and incubated overnight at 37 °C. The development of a blue color in senescent cells was observed using a microscope, and the stained cells were counted under light microscopy (200×). The percentage of blue-stained cells was calculated by dividing the number of blue-stained cells by the total number of cells in the field.

### 2.12. Measurement of Cellular Iron Levels 

An iron assay kit (cat. no. 8448, ScienCell, Carlsbad, CA, USA) was used to detect total iron concentration in the cell lysates and kidney tissues according to the manufacturer’s instructions. The renal cells and tissues were homogenized in four volumes of iron assay buffer and then centrifuged at 13,000× *g* for 10 min to eliminate insoluble materials. Assay buffer, reductant, and sample reagent were added into supernatants to test total iron (Fe^3+^ plus Fe^2+^) levels. The absorbance at a wavelength of 590 nm was measured on a microplate reader after 30 min of incubation at 37 °C.

### 2.13. Detection of Fe^2+^ Using Fluorescent Probes

Intracellular Fe^2+^ levels were assessed using FerroOrange probes (F374, Dojindo Kumamoto, Japan). Following the manufacturer’s protocol, the cells were treated with IS and incubated for a duration of 24 h. Then, the supernatant was discarded, and the cells were washed with a serum-free medium. The cells were then treated with 1 μM FerroOrange working solution for 30 minutes at 37 °C. Subsequently, a confocal microscope (LSM 880, Carl Zeiss, Jena, Germany) equipped with a 63× oil immersion objective was used to visualize the Fe^2+^ fluorescence. 

### 2.14. Lipid Peroxidation Assay 

The intracellular MDA contents were measured using a commercial TBRAS Assay Kit (cat. no. 10009055, Cayman); MDA was as standard. Briefly, the cells (2 × 10^7^) were collected in 1 mL of PBS and sonicated on ice. Sample, SDS solution, and color reagent were then added to each vial and boiled for 1 h. After incubation on ice for 10 min, the samples were centrifuged at 1600× *g* for 10 min. After a stable incubation at room temperature for 30 min, the absorbance was measured at 530–540 nm.

### 2.15. Total and Reduced Glutathione (GSH) Measurement 

A Glutathione Assay kit (cat. no. CS0260, Sigma-Aldrich) was used to measure the intracellular levels of total glutathione (GSSG and GSH). The cells were deproteinized with a 5% solution of 5-sulfosalicylic acid, and then the precipitated proteins were removed by centrifuge. A kinetic assay was used to detect the GSH levels with which catalytic amounts (nmoles) of GSH caused a continuous reduction of 5,5′-dithiobis (2-nitrobenzoic acid) (DTNB) to TNB, and the formed GSSG was recycled via glutathione reductase and NADPH. A microplate reader was used to determine the absorbance at 412 nm.

### 2.16. Statistical Analysis

For the in vitro model, a one-way ANOVA with Tukey’s post hoc test was used to analyze the significant difference among multi-groups, and the data are presented as means  ±  SD. For the in vivo model, data were presented as median (minima and maxima). The Kruskal–Wallis test was used to analyze statistical differences. All statistical analyses were performed using Prism software (version 7.0) by GraphPad. Significance was defined at a *p*-value threshold of <0.05

## 3. Results

### 3.1. Indoxyl Sulfate (IS) Induces Senescence through ROS in Renal Tubular Cells

We first examined the cell viability of NRK-52E cells after exposure to IS for 24 h and found that cell viability was decreased with treatment with 0.5 and 1 mM IS (Figure 1A). To investigate whether IS induces senescence in renal tubular cells, we determined senescence biomarkers including p53, p21, and p16 protein expressions as well as senescence-associated beta-galactosidase (SA-β-gal) staining after treatment with IS (0.1–1 mM) in NRK-52E cells. Significantly elevated protein expression levels of p53, p21, and p16 were observed in cells exposed to 0.5 mM IS (Figure 1B), and the SA-β-gal activity was also increased with IS treatments (Figure 1C). Furthermore, ROS generation was significantly increased due to IS, which could be effectively inhibited by the antioxidant N-acetyl-l-cysteine (NAC) (Figure 1D). We further examined whether NAC could alleviate IS-induced senescence. As shown in Figure 1E,F, NAC treatment substantially decreased the heightened protein expression levels of p53, p21, and p16, along with a reduction in SA-β-gal activity that was induced by IS. These results suggest that ROS participates in the IS-induced senescence in renal tubular cells. 

### 3.2. IS Triggers Renal Fibrosis/Injury via ER Stress, Ferroptosis, and Epithelial–Mesenchymal Transition (EMT) 

ER stress, ferroptosis, and EMT have been reported to be involved in IS-related diseases [23,24,25]. We hypothesize that these factors may also participate in renal tubular damage caused by IS. To determine the effect of IS on the progression of renal injury, we examined the protein expression of key biomarkers associated with ER stress, including GRP78, p-eIF2α, ATF4, ATF6, and CHOP. Interestingly, our findings indicated an elevation in ATF4 and CHOP protein expression following treatment with 0.5 and 1 mM of IS. However, no significant alterations were observed in the expressions of GRP78, p-eIF2α, and ATF6 (Figure 2A). Then, we determined the protein expressions of GPX4, PTGS2, and xCT, which are the important markers of ferroptosis. Remarkably, while PTGS2 expression increased, the expression of GPX4 diminished in cells treated with 0.5 and 1 mM IS compared with the control group. However, xCT protein expression remained relatively unchanged (Figure 2B). IS-induced ER stress and ferroptosis were effectively inhibited by NAC treatment (Figure 2C). Previous studies have demonstrated that senescence, ER stress, and ferroptosis are associated with the development and progression of renal fibrosis [26,27,28]. Accordingly, our focus shifted to assessing whether IS stimulated the EMT process, thereby causing fibrosis in renal tubular cells. We also found that the levels of fibronectin and vimentin protein expression were significantly increased and the E-cadherin protein expression was significantly reduced in IS-treated cells. These effects were effectively reversed by the antioxidant NAC or iron chelator DFO treatment (Figure 2D). These results indicated that IS induced ER stress (via ATF4-CHOP axis), ferroptosis, and EMT through ROS stimulation in renal tubular cells.

### 3.3. IS Induces the Accumulation of Intracellular Iron and Causes Ferroptosis in Renal Tubular Cells

Previous studies have demonstrated that high levels of labile iron and ferroptosis are the main drivers of ROS generation and lipid peroxidation. Thus, we investigated the role of iron metabolism in IS-induced ferroptosis and renal injury. To unravel this intricate web, we turned our attention to key players in cellular iron homeostasis. We investigated the levels of key proteins including transferrin (Tf), transferrin receptor (TfR), ferritin (FTH), divalent metal transporter 1 (DMT1), ferroportin (FPN), NCOA4, and heme oxygenase-1 (HO-1) in NRK-52E cells subjected to IS treatment. As shown in Figure 3A, IS-treated cells exhibited significantly elevated protein expression levels of Tf, TfR, and FTH. FPN showed a significant induction in the protein expression in cells treated with 1 mM IS. Conversely, the protein expression of DMT1 and NCOA4 did not exhibit significant changes in IS-treated cells compared with the control cells. Furthermore, IS treatment resulted in a significant increase in the protein expression of HO-1 (Figure 3A), a critical enzyme involved in ferroptosis [29,30]. 

There are other characteristics of ferroptosis, including lipid peroxidation, increase in ROS levels, accumulation of intracellular iron, and decrease in glutathione (GSH) levels [31]. As shown in Figure 3B, IS treatment induced a significant increase in the levels of malondialdehyde (MDA), which is a well-known secondary product of lipid peroxidation and is often used as a marker of cellular injury [32]. Our findings indicate that the ROS levels were significantly increased in IS-treated cells (Figure 1D and Figure 3C). In line with these findings, IS treatment correlated with significant increases in intracellular total iron contents (Figure 3D) and Fe^2+^ levels (Figure 3E). Conversely, GSH levels exhibited a significant reduction in renal tubular cells upon IS treatment (Figure 3F). 

To further explore the potential role of iron accumulation in IS-induced renal injury, we examined whether DFO could alleviate IS-induced adverse effects including oxidative stress, senescence, ER stress, and ferroptosis through downregulation of intracellular iron levels in renal tubular cells. In our experimental design, we subjected cells to pretreatment with DFO (50 µM) for 1 h, followed by subsequent treatment with IS (at concentrations of 0.5 and 1 mM) for 24 h. As shown in Figure 3, DFO treatment significantly reversed the effects induced by IS. These effects included the increased levels of MDA (Figure 3B), ROS (Figure 3C), intracellular iron accumulation (Figure 3D,E), and the decrease in GSH levels (Figure 3F). DFO treatment significantly inhibited the increased expression of senescence-related p53, p21, and p16 proteins in response to 0.5 mM IS treatment (Figure 4A). Consistently, the increased SA-β-gal activity induced by IS was also notably curtailed upon DFO treatment (Figure 4B). The increased expression of ER stress-related ATF4 and CHOP proteins induced by IS was also dramatically attenuated after DFO pretreatment (Figure 4C). Additionally, the decreased GPX4 protein expression and the increased PTGS2 protein expression induced by IS were both significantly restored following DFO pretreatment (Figure 4D). Furthermore, DFO treatment emerged as a potent modulator, significantly reducing the increased protein expression levels of TfR and FTH induced by IS in renal tubular cells (Figure 4E). Collectively, these findings suggest that IS induces senescence, ER stress, and ferroptosis in renal tubular cells through its promotion of iron accumulation. Therefore, DFO may be a potential therapeutic approach to prevent IS-induced kidney damage.

### 3.4. IS Accumulation Causes Renal Injury and Senescence in Adenine-Induced CKD Mice

To investigate the potential pathological role of IS in the kidneys of adenine-induced CKD mice, we assessed renal injury using a serum biochemistry test, hematoxylin and eosin (H&E), periodic acid Schiff (PAS), and Masson’s trichrome staining. Obvious increases in serum creatinine (SCr), BUN, and IS levels were observed in the adenine-induced CKD mice and were significantly ameliorated by treatment with AST-120, an adsorbent charcoal specifically designed to scavenge the IS precursor in the intestine (Figure 5A; Table 1). Several facets of renal injuries, such as brush border loss, obvious tubular dilation, excessive total collagen deposition (detected via Masson’s trichrome staining), and PAS-positive deposits in the kidneys of adenine-induced CKD mice were observed. AST-120 treatment exhibited a commendable capacity to ameliorate these renal injuries (Figure 5B). We next investigated the senescence-related signals in the kidneys of the adenine-induced CKD model and tested whether AST-120 could exert an effect on senescence. It was found that the expression levels of p53, p21, p16, and SA-β-gal proteins were significantly increased in the CKD kidneys. However, these increases were significantly reversed by AST-120 treatment (Figure 5D). These findings suggest that AST-120 alleviates renal injury and senescence by inhibiting IS synthesis in adenine-induced CKD mice.

### 3.5. IS Contributes to ER Stress, Ferroptosis, Iron Accumulation, Nrf2/HO-1 Activation, and the EMT Process in CKD Kidneys

We further verified the involvement of IS in the induction of key processes, including ER stress, ferroptosis, iron accumulation, Nrf2/HO-1 activation, and EMT in the kidneys of adenine-induced CKD mice with or without AST-120 treatment. We found that the expression of GRP78, ATF4, and CHOP proteins significantly increased in the CKD kidneys. Notably, AST-120 treatment significantly attenuated the increased expression of ATF4 and CHOP proteins (Figure 6A). These results suggest that IS contributes to the ER stress induction via the ATF4-CHOP axis in the CKD kidneys. 

Proceeding to the exploration of ferroptosis, our observations unveiled significant reductions in the expression levels of GPX4 and xCT proteins, coupled with a noteworthy increase in PTGS2 protein expression in the CKD kidneys (Figure 6B). These effects in the CKD kidneys were counteracted by AST-120 treatment (Figure 6B). We also observed an increase in the 4-hydroxynonenal (4-HNE; a product of lipid peroxidation) expression and a decrease in the GPX4 expression based on IHC staining analysis (Figure 6C). Moreover, we noted elevated iron deposition (Figure 6D) and elevated total iron content (Figure 6E; Table 1) in the kidneys of CKD mice. AST-120 treatment effectively reversed these trends (Figure 6D,E). These data suggest that IS induces iron accumulation and ferroptosis in the CKD kidneys.

We next investigated the iron metabolism in the kidneys of adenine-induced CKD mice. As shown in Figure 7A, the upregulation of key proteins including Tf, TfR, FTH, DMT1, and FPN, but not NCOA4 protein expressions was observed in the CKD kidneys. The increased FTH protein expression determined via IHC staining was also observed in the CKD kidneys (Figure 7B). Of note, treatment with AST-120 effectively reversed the elevated protein expression of Tf, TfR, FTH, DMT1, and FPN (Figure 7A,B). The Nrf2/HO-1 signaling plays an important role in the intracellular defense system against oxidative stress [33] and is involved in ferroptosis [30,34]. As shown in Figure 7C, the levels of Nrf2, Keap-1, and HO-1 protein expression were significantly enhanced in the CKD kidneys. The increased HO-1 protein expression was also observed in the CKD kidneys, as determined via IHC staining (Figure 7D). AST-120 treatment effectively reduced the increase in Nrf2, Keap-1, and HO-1 protein expression (Figure 7C,D). Furthermore, we examined the EMT process in the kidneys of adenine-induced CKD mice. As shown in Figure 7E, the fibronectin and vimentin protein expression levels were significantly enhanced, and the E-cadherin protein expression levels were significantly reduced in the CKD kidneys compared with the control kidneys. Treatment with AST-120 significantly attenuated the EMT process in the CKD kidneys (Figure 7E). These results suggest that IS causes iron accumulation and activates the Nrf2/HO-1 signaling pathway as well as the EMT process to drive renal fibrosis and injury in adenine-induced CKD mice.

## 4. Discussion

Studies have shown that IS generates ROS, causes oxidative stress, and promotes renal injury, inflammation, and fibrosis in various cell types, including renal tubular cells, endothelial cells, and vascular smooth muscle cells [35,36,37,38,39]. Although previous studies have shown that IS activates the NADPH oxidase system and leads to the production of superoxide anion (O^2•−^) and other ROS [40,41], the mechanism by which IS generates ROS is not fully understood. Mitochondria produce hydrogen peroxide through respiration, which is then converted into harmless H_2_O by catalase or other enzymes. However, excess hydrogen peroxide undergoes redox reactions with iron in the body via the Fenton reaction, leading to the production of more free radicals, oxidative stress, and lipid peroxidation [42]. Here, we found that the Tf and TfR protein expressions were higher after IS treatment in vitro and in CKD mice, which were primarily responsible for facilitating cellular uptake of iron bound to transferrin (Fe^3+^-transferrin). Another noteworthy finding was the enhanced expression of FTH, the main cellular iron storage protein. This heightened FTH expression suggests an augmentation in cellular iron accumulation due to IS exposure and CKD in mice. This intricate process results in excessive ROS production and lipid peroxidation, with a potential contribution from the Fenton reaction. Moreover, the administration of the iron chelator DFO was able to reverse the adverse effects induced by IS. These findings provide insight into the mechanism by which IS induces oxidative stress, consequently shedding light on potential therapeutic targets for CKD. Furthermore, FPN, which exhibited iron efflux from the cells under certain conditions, showed a significant induction in the protein expression. Moreover, the results indicate that DMT1, which is responsible for transporting iron (Fe^2+^) from endosomes to cytoplasm, was significantly increased in CKD mice but did not exhibit significant changes in IS-treated cells, indicating that the mechanism of iron metabolism is still somewhat different in vitro and in vivo. NCOA4, which is the receptor for ferritinophagy, triggers autophagic degradation of ferritin and releases iron as free iron, did not exhibit significant changes [30,31]. However, the NCOA4 protein expression was not significantly changed in both IS-treated renal cells and CKD mouse kidneys. These results indicate that ferritinophagy may be less involved in IS-related renal injury.

Previous studies have suggested an association between iron deposition and renal tubular injury in various types of CKD, as elevated urinary iron levels and renal iron deposition have been found in these patients [43,44]. While emerging evidence demonstrates a link between ferroptosis and kidney injury [45,46], the regulatory mechanisms of ferroptosis in CKD remain unclear. In the present study, we utilized an adenine-induced CKD model with AST-120 treatment to identify the pivotal role of IS as a crucial contributor to ferroptosis in CKD. The underlying mechanism of IS-induced ferroptosis can be attributed to the augmentation of iron uptake, facilitated through the action of transferrin and transferrin receptors. Interestingly, a previous study found that 3-Carboxy-4-methyl-5-propyl-2-furanpropanoic acid (CMPF), another protein-bound uremic toxin, induces ferroptosis in HK-2 cells and NRK49F cells [47]. This study revealed intriguingly divergent findings compared with our own. Specifically, CMPF treatment led to a significant decrease in the protein expression of GPX4 and ferritin, while markedly increasing the expression of TfR [47]. This contrast suggests that distinct uremic toxins may indeed exert diverse influences on iron metabolism.

Iron is an essential component in human physiology and plays a vital role in various proteins such as hemoglobin, myoglobin, and cytochrome [48,49]. However, an excessive accumulation of iron in the kidneys has been associated with both acute kidney injury (AKI) and CKD. Patients with kidney disease often experience high concentrations of iron within their renal tubules due to the increased filtration of iron and iron-containing proteins, such as hemoglobin, transferrin, and neutrophil gelatinase-associated lipocalin (NGAL) in the glomeruli [50,51,52,53]. Therefore, one of the potential therapy strategies for AKI and CKD is to decrease the elevated luminal or intracellular iron levels in the kidneys [53]. DFO, an agent approved by the FDA for the treatment of iron overload [54], has emerged as a promising intervention. Previous studies have shown that DFO attenuates aortic calcification in rats with 5/6 nephrectomy-induced CKD [55] and prevents renal tubulointerstitial fibrosis in a unilateral ureteral obstruction (UUO)-CKD mouse model [56]. In our present study, we found that DFO pretreatment effectively attenuated iron accumulation and prevented ferroptosis, senescence, ER stress, and fibrosis in renal tubular cells treated with IS. Iron accumulation, ferroptosis, senescence, ER stress, and fibrosis also appeared in the kidneys of adenine-induced CKD mice. AST-120 treatment reversed these adverse changes in the kidneys. These findings suggest that IS-induced iron accumulation and ferroptosis may play an important role in CKD progression.

ER stress has been demonstrated to promote ferroptosis through the degradation of FTH, a process known as ferritinophagy [24], whereas excess iron accumulation triggers the unfolded protein response and leads to ER stress [57]. To clarify the relationship between ER stress and iron accumulation induced by IS in renal cells in vitro and in vivo, our study showed that DFO treatment effectively attenuated ER stress by downregulating iron levels. Furthermore, senescent cells tend to accumulate large amounts of intracellular iron regardless of stimuli, such as irradiation, replication, or oncogenesis [58]. Our study found that IS-induced senescence was inhibited by DFO in the cultured renal tubular cells and the CKD kidneys. However, the concentration of 1 mM of IS could not induce senescence in the cultured renal tubular cells in our study. While the actual cause is unknown, an alternative interpretation might be that this concentration of IS potentially induced cell death, conceivably through the mechanism of ferroptosis. Moreover, our findings showed that the Keap-1/Nrf2/HO-1 signaling pathway was induced both in vitro and in vivo. The Nrf2/HO-1 signaling pathway has been extensively studied as a key regulatory mechanism for intracellular defense against oxidative stress [33,59]. However, the role of the Nrf2/HO-1 pathway in the realm of ferroptosis is notably complex, exhibiting seemingly paradoxical pro-ferroptotic and anti-ferroptotic properties in various in vitro and in vivo models [29,30,60,61]. The role of the Nrf2/HO-1 pathway in IS-related ferroptosis and renal injury remains to be fully clarified.

Renal fibrosis accompanied by excessive accumulation and deposition of the extracellular matrix (ECM) component is a prevalent pathological characteristic in CKD that has been characterized [62,63]. A previous study has shown that iron accumulation was associated with renal interstitial fibrosis in the kidney of a rat CKD model [21]. Additionally, it is generally believed that ER stress, senescence, and ferroptosis result in the development of renal fibrosis [26,27,28]. In this study, utilizing an IS-treated renal tubular cell model and an adenine-induced CKD mouse model with AST-120 treatment, we provided evidence that IS may directly induce senescence, ER stress, and ferroptosis through the accumulation of iron in renal cells, leading to the development of renal fibrosis.

AST-120 is an oral activated carbon adsorbent that is widely used as a therapeutic to inhibit circulating uremic toxins in CKD patients [64]; it provides renoprotection by preventing glomerular hypertrophy, fibrosis, and proteinuria, and delays the initiation of dialysis in progressive CKD [35,65]. In addition, AST-120 also improves bone health, preventing uremic osteoporosis by maintaining bone mass and reducing susceptibility to fractures [66,67,68]. Moreover, AST-120 addresses CKD-related inflammation by inhibiting immune cell activation, oxidative stress, and inflammatory responses [69,70]. It also protects endothelial function, enhances vasodilation, and improves vascular health [71]. Notably, recent studies have shown that AST-120 exhibits potential cardioprotective effects by reducing aortic IS deposition, attenuating atherosclerosis, and protecting the heart and kidneys [72]. In the gastrointestinal environment, AST-120 restores mucosal integrity, corrects microbial imbalances, and alleviates renal decline [73,74]. In summary, AST-120 has wide therapeutic effects. In the present study, we found that AST-120 can reduce the accumulation of iron in the kidneys of CKD mice and significantly improve kidney injury and fibrosis caused by senescence, ER stress, and ferroptosis, which reconfirmed the therapeutic potential of AST-120 in the treatment of CKD.

Several limitations of this study should be considered. First, we utilized a cultured renal cell model and an adenine-induced CKD mouse model to investigate the potential pathological role of IS. IS is a metabolite formed in the body through the breakdown of dietary tryptophan by gut bacteria and is subsequently processed by the liver. However, it is important to note that these models may not directly induce changes in gut microbial metabolism or affect the production of metabolites like IS. Second, the in vivo role of iron in CKD mice is not clarified. Third, the information about the relationship between IS and ferroptosis in CKD patients is lacking. We may consider conducting experiments to analyze these issues in the future.

## 5. Conclusions

In conclusion, our results suggest that uremic toxin-IS is an instigator of iron accumulation, ROS generation, ferroptosis, senescence, and ER stress, thereby driving the development of renal injury and fibrosis in cultured renal tubular cells and mouse CKD kidneys, which could be effectively alleviated by treatments with an iron chelator, deferoxamine (DFO), in vitro and an adsorbent charcoal, AST-120 (scavenging the IS precursor), in vivo (Figure 8). These findings reveal a deeper insight into the mechanisms underlying IS-related renal toxicity and provide potential therapeutic strategies for preventing or treating CKD. By identifying iron accumulation as a central regulator of IS-induced adverse effects on renal cells, our study emphasizes the importance of monitoring and controlling iron and IS levels in CKD patients to prevent or mitigate renal injury. 

## Figures and Tables

**Figure 1 antioxidants-12-01931-f001:**
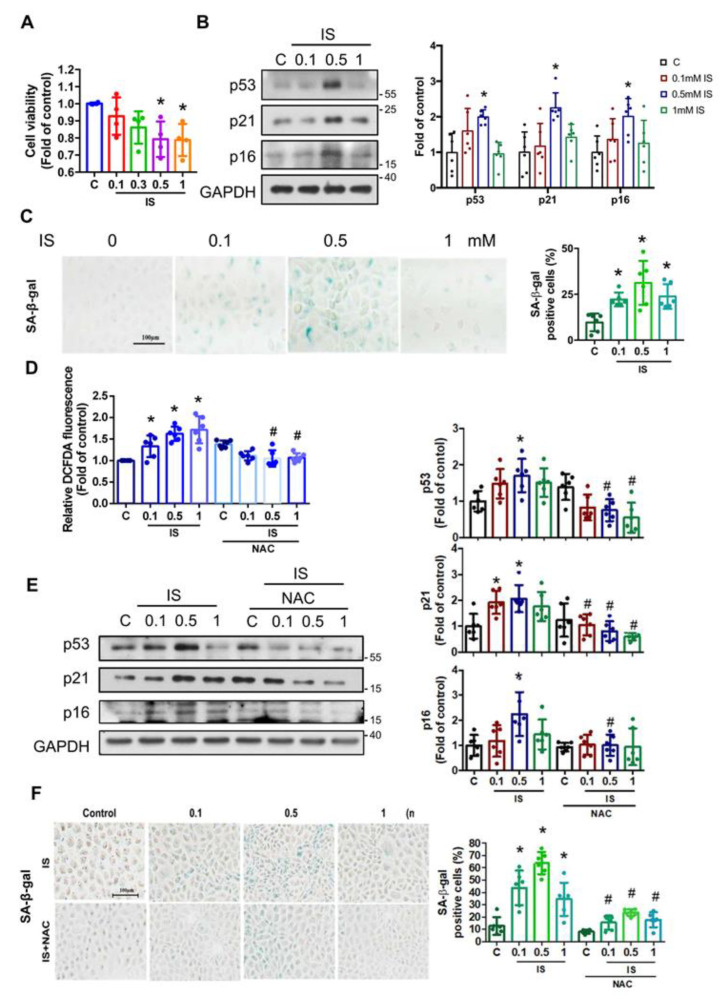
Indoxyl sulfate (IS) induced senescence through reactive oxygen species (ROS) signaling in the renal tubular cells. NRK-52E cells were treated with varying doses of IS for 24 h with or without pretreatment of 1 mM N-acetylcysteine (NAC) for 1 h. (**A**) Cells were incubated for 24 h in the presence of IS (0.1–1 mM), and the cell viability was measured using an MTT assay. (**B**) Western blot analysis and quantification for p53, p21, and p16 protein expressions. (**C**) The senescence-associated beta-galactosidase (SA-β-gal) staining and quantification. Scale bar = 100 μm. (**D**) The ROS production in IS-treated NRK-52E cells with or without NAC. (**E**) Western blot analysis and quantification for p53, p21, and p16 protein expressions in the IS-treated NRK-52E cells with or without NAC. (**F**) The SA-β-gal staining and quantification in the IS-treated NRK-52E cells with or without NAC. Scale bar = 100 μm. Data are presented as means ± SD (*n* = 4 in (**A**), *n* = 6 in (**B**–**F**); independent experiments). *, *p*  <  0.05, compared with the control (**C**) group. #, *p*  <  0.05, compared with the corresponding concentrations of IS groups. One-way ANOVA followed by Tukey’s post hoc test was used to analyze statistical differences.

**Figure 2 antioxidants-12-01931-f002:**
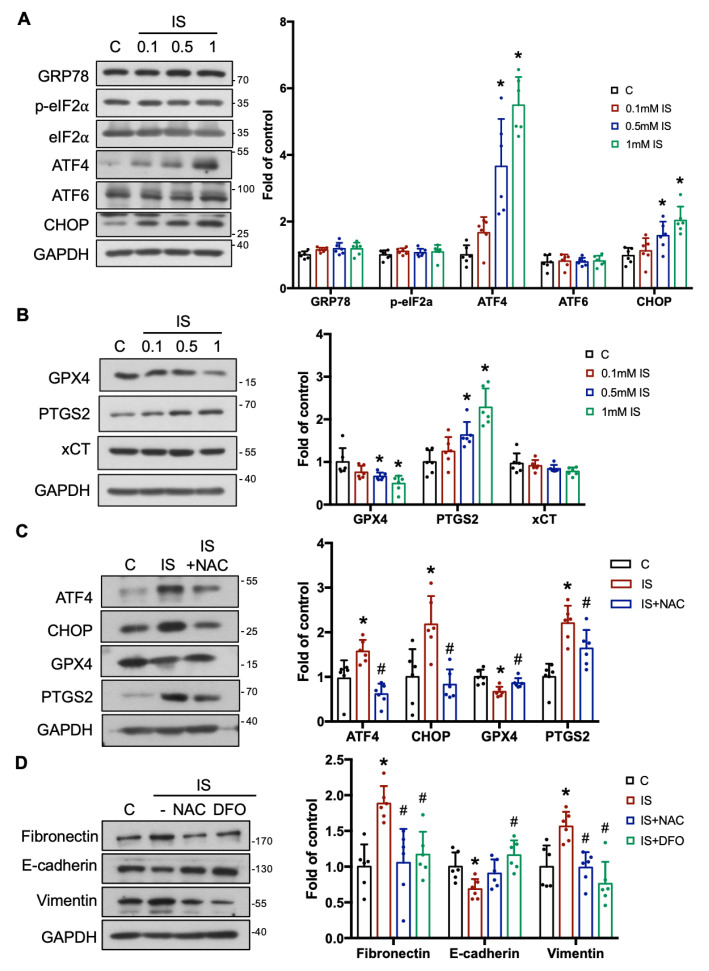
IS induced ER stress, ferroptosis, and epithelial–mesenchymal transition (EMT) through ROS signaling in the renal tubular cells. NRK-52E cells were treated with various concentrations of IS (0.1–1 mM) for 24 h with or without pretreatment of 1 mM NAC for 1 h or 50 µM deferoxamine (DFO) for 1 h. (**A**,**B**) Western blot analysis and quantification for ER stress-related GRP78, p-eIF2α, ATF4, ATF6, and CHOP (**A**), and ferroptosis-related GPX4, PTGS2, and xCT (**B**) protein expression. (**C**) Western blot analysis and quantification for ATF4, CHOP, GPX4, and PTGS2 protein expression in IS-treated NRK-52E cells with or without NAC. (**D**) Western blot analysis and quantification for EMT-related fibronectin, E-cadherin, and vimentin protein expression in the IS-treated NRK-52E cells with or without NAC or DFO. Data are presented as means ± SD (*n*  =  6 in (**A**–**D**); independent experiments). *, *p*  <  0.05, compared with the control (**C**) group. #, *p*  <  0.05, compared with the corresponding concentrations of IS groups. One-way ANOVA followed by Tukey’s post hoc test was used to analyze statistical differences.

**Figure 3 antioxidants-12-01931-f003:**
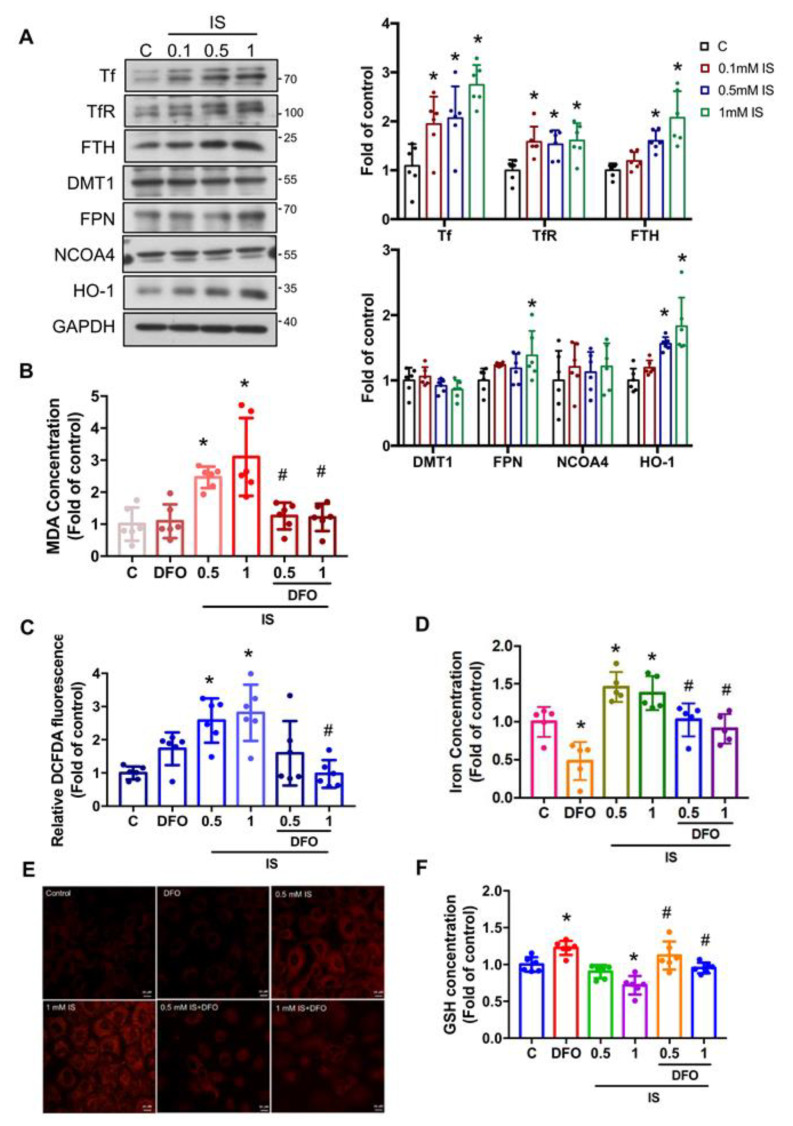
IS altered iron metabolism and induced lipid peroxidation, ROS production, and iron accumulation in the renal tubular cells. NRK-52E cells were treated with various concentrations of IS (0.1–1 mM) for 24 h with or without pretreatment of 50 µM of DFO for 1 h. (**A**) Western blot analysis and quantification for Tf, TfR, FTH, DMT1, FPN, NCOA4, and HO-1 protein expression. (**B**–**E**) The levels of malondialdehyde (MDA) (**B**), ROS production (**C**), total iron (**D**), fluorescent ferrous ion levels assessed via confocal microscopy (**E**), and total GSH (**F**) in the IS-treated NRK-52E cells with or without DFO. Total iron levels were detected via an iron assay kit. Intracellular Fe^2+^ levels were detected using FerroOrange assay, and images are shown with confocal microscopy images; scale bar = 10 μm. Data are presented as means ± SD (*n* = 5 in (**D**), *n*  =  6 in (**A**–**C**,**E**,**F**); independent experiments). *, *p*  <  0.05, compared with the control (**C**) group. #, *p*  <  0.05, compared with the corresponding concentrations of IS groups. One-way ANOVA followed by Tukey’s post hoc test was used to analyze statistical differences.

**Figure 4 antioxidants-12-01931-f004:**
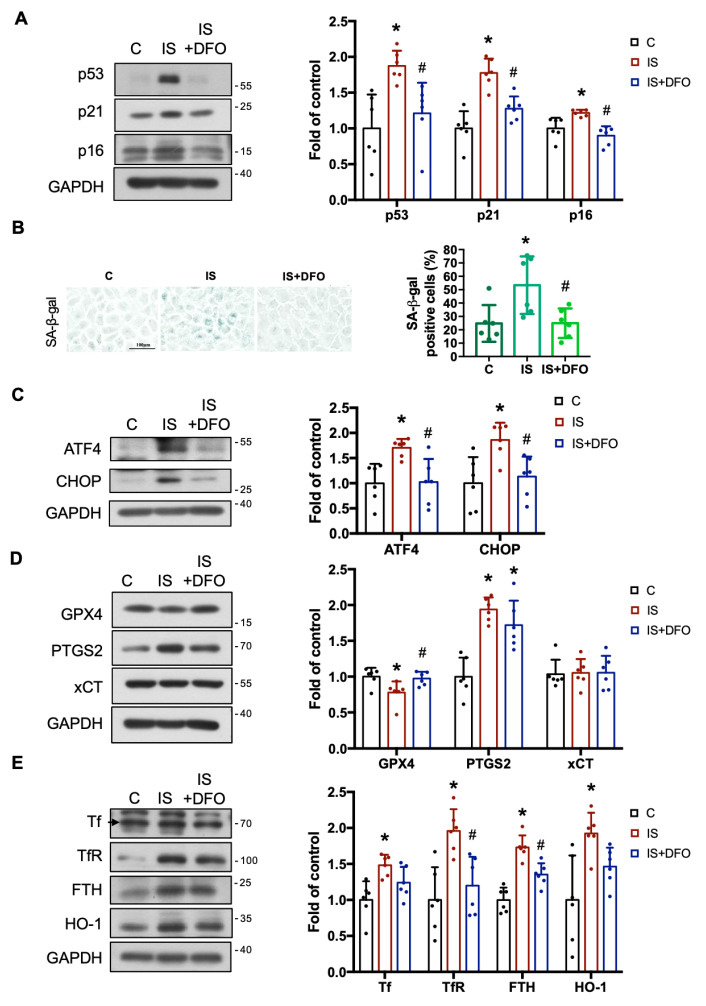
DFO attenuated IS-induced senescence, ER stress, ferroptosis, and iron metabolism alteration in renal tubular cells. NRK-52E cells were treated with 0.5 mM of IS for 24 h with or without pretreatment of 50 µM of DFO for 1 h. (**A**) Western blot analysis and quantification for p53, p21, and p16 protein expressions. (**B**) The SA-β-gal staining and quantification; scale bar = 100 μm. (**C**–**E**) Western blot analysis and quantification for ATF4 and CHOP (**C**), GPX4, PTGS2 and xCT (**D**), and Tf, TfR, FTH, and HO-1 (**E**) protein expressions. Data are presented as means ± SD (*n* = 5 in (**D**), *n*  =  6 in (**A**–**C**,**E**); independent experiments). *, *p*  <  0.05, compared with the control (**C**) group. #, *p*  <  0.05, compared with the IS group. One-way ANOVA followed by Tukey’s post hoc test was used to analyze statistical differences.

**Figure 5 antioxidants-12-01931-f005:**
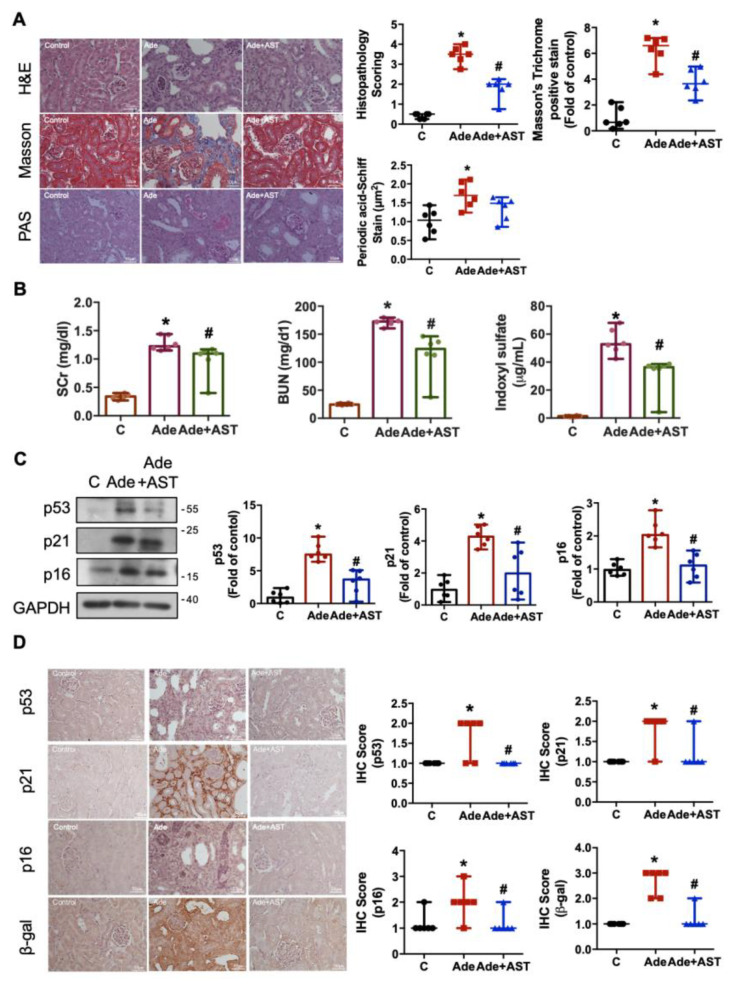
The alterations in the renal function, fibrosis, and senescence in the adenine-induced CKD mice, which could be attenuated by AST-120. (**A**) The changes in serum creatinine, BUN, and IS levels in CKD mice. (**B**) The stains and quantification for H&E, Masson’s trichrome, and PAS staining in the kidneys; scale bar = 50 μm. (**C**) Western blot analysis and quantification for p53, p21, and p16 protein expressions in the kidneys. (**D**) The IHC staining and quantification for senescence-related p53, p21, p16, and β-galactosidase (β-gal) in the kidneys; scale bar = 50 μm. Data are presented as median (minima and maxima) (*n* = 6 mice/group). *, *p*  <  0.05, compared with the control (**C**) group. #, *p * <  0.05, compared with the adenine (Ade) group. The Kruskal–Wallis test was used to analyze statistical differences.

**Figure 6 antioxidants-12-01931-f006:**
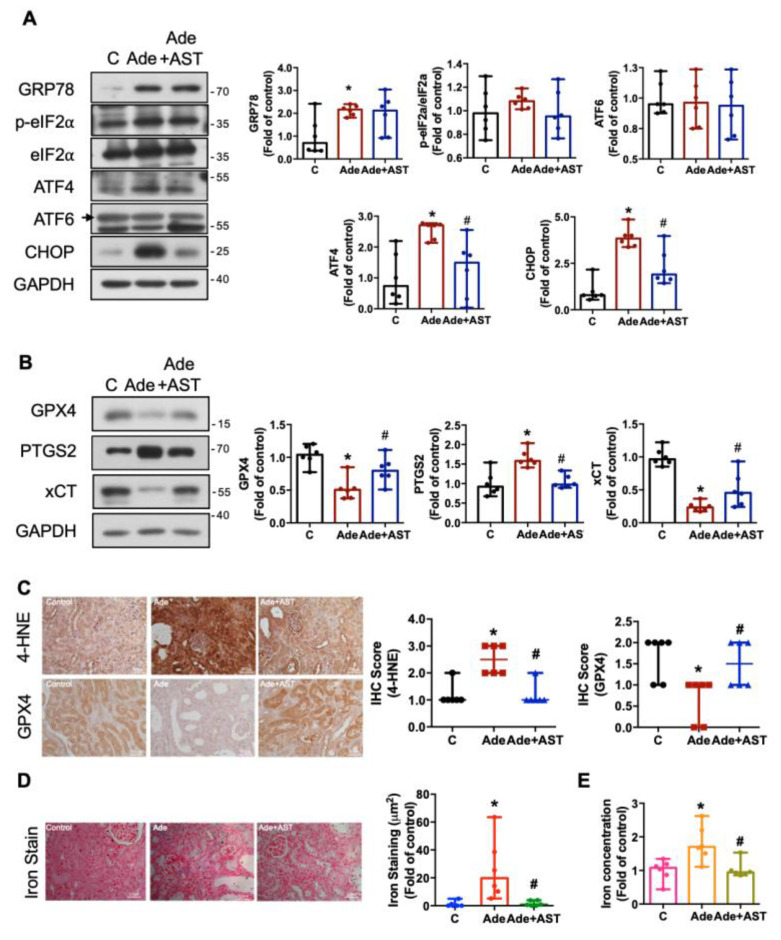
The induction of ER stress, ferroptosis, lipid peroxidation, and iron accumulation in the kidneys of CKD mice, which could be attenuated by AST-120. (**A**,**B**) Western blot analysis and quantification for GRP78, p-eIF2α, ATF4, ATF6, and CHOP (A) and GPX4, PTGS2, and xCT (**B**) protein expression in the kidneys. (**C**) The IHC staining and quantification for lipid peroxidation end-product 4-HNE and ferroptosis marker GPX4 in the kidneys; scale bars = 50 μm. (**D**) The iron accumulations in the kidneys determined using an iron stain (blue color) with a neutral fast red counterstain to distinguish between nuclei (red) and cytoplasm (pink); scale bar = 50 μm. (**E**) Total iron contents in the kidneys determined using an iron assay kit. Data are presented as median (minima and maxima) (*n* = 6 mice/group). *, *p*  <  0.05, compared with the control (**C**) group. #, *p*  <  0.05, compared with the adenine (Ade) group. The Kruskal–Wallis test was used to analyze statistical differences.

**Figure 7 antioxidants-12-01931-f007:**
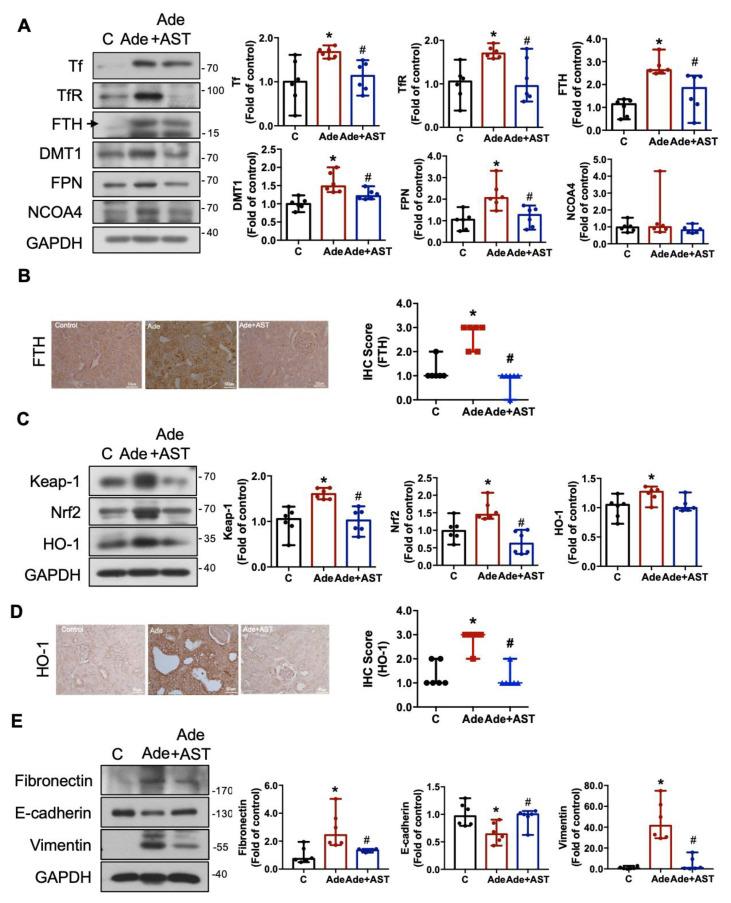
The alterations in iron metabolism, Nrf2/HO-1 signaling, and EMT process in the kidneys of CKD mice, which could be attenuated by AST-120. (**A**) Western blot analysis and quantification for iron metabolism-related Tf, TfR, FTH, DMT1, FPN, and NCOA4 protein expression in the kidneys. (**B**) The IHC staining and quantification for FTH in the kidneys; scale bar = 50 μm. (**C**) Western blot analysis and quantification for intracellular antioxidant system-related Keap-1, Nrf2, and HO-1protein expression in the kidneys. (**D**) The IHC staining and quantification for HO-1 in the kidneys; scale bar = 50 μm. (**E**) Western blot analysis and quantification for EMT-related fibronectin, E-cadherin, and vimentin protein expression in the kidneys. Data are presented as median (minima and maxima) (*n* = 6 mice/group). *, *p*  <  0.05, compared with the control (**C**) group. #, *p*  <  0.05, compared with the adenine (Ade) group. The Kruskal–Wallis test was used to analyze statistical differences.

**Figure 8 antioxidants-12-01931-f008:**
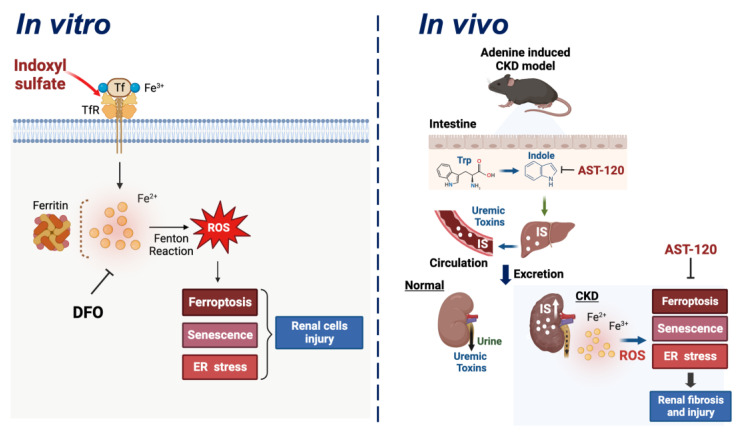
A schematic model illustrating that indoxyl sulfate (IS) alters iron metabolism, triggers iron accumulation, and elevates ROS levels by the Fenton reaction, leading to the development of renal injury and fibrosis in cultured renal tubular cells and mouse CKD kidneys through ferroptosis, senescence, and ER stress signaling pathways, which can be rescued by treatments with an iron chelator, deferoxamine (DFO), in vitro and an adsorbent charcoal, AST-120 (scavenging the IS precursor), in vivo. Created with BioRender.com (accessed on 23 September 2023).

**Table 1 antioxidants-12-01931-t001:** Biochemical indicators and total iron contents among different groups in the adenine mouse model.

	Control	Adenine	Adenine + AST120
Serum BUN (mg/d1)	24.6 (23.3, 27.2)	172.8 (160.4, 179.7) *	123.7 (37.6, 146.3) *^,#^
Serum creatinine (mg/dl)	0.34 (0.27, 0.4)	1.22 (1.15, 1.44) *	1.09 (0.4, 1.17) *^,#^
Serum indoxyl (μg/mL)	1.33 (0.75, 1.88)	52.8 (42.31, 68.05) *	36.34 (4.27, 38.6) *^,#^
Renal total iron content (μM)	0.89 (0.36, 1.11)	1.41 (0.92, 2.17) *	0.78 (0.71, 1.27) ^#^

Data are presented as median (minima and maxima) (*n* = 6 mice/group). The Kruskal–Wallis test was used to analyze statistical differences. *, *p*  <  0.05, compared with the control group. ^#^, *p*  <  0.05, compared with the adenine group.

## Data Availability

The data presented in this study are available from the corresponding author upon reasonable request.

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
