# Peer review of "Inhibition of Indoxyl Sulfate-Induced Reactive Oxygen Species-Related Ferroptosis Alleviates Renal Cell Injury In Vitro and Chronic Kidney Disease Progression In Vivo"

_antioxidants, 2023, doi:10.3390/antiox12111931_

Round 1
Reviewer 1 Report
Comments and Suggestions for Authors
The manuscript entitled “Inhibition of Indoxyl Sulfate-Induced Reactive Oxygen Species-Related Ferroptosis Alleviates Renal Cell Injury In Vitro and Chronic Kidney Disease Progression In Vivo” explores whether indoxyl sulfate induces ferroptosis and injury, and whether it affects iron metabolism in the renal cells and the kidneys. It is an interesting investigation and very convincing in that it includes cell cultures and also experimental animals. The data appear very interesting although this work can be further strengthened by addressing the following comments:
Major
1) The scope section (end paragraph of introduction) somehow predefines the upcoming findings and the there from emerging discussion points. This is somehow confusing, especially expressions such as “we verified” and “the findings … illustrated” don’t belong in the Introduction.
2) In the Materials and Methods sub-section 2.3 the groups of animals are defined imprecisely. The addition of a sub-section entitled experimental design one for the cell cultures experiment and one for the animal experiments would add clarity for the reader.
3) The Results section includes not only the findings but also a lot of discussion and controversies about the findings. There are for example too many references that are part of the discussion of the findings. If this text and these references were added in the discussion it would make the discussion more extensive and would make the paper more clear cut.
4) For example the missing effect on NCOA4 both in RTC and in CKD mice is mentioned in the findings and discussed in a concise form there but is not even mentioned in the discussion section.
5) With a sample size of 6 mice per group, the authors should switch to non-parametric statistical analysis. Towards this end, please report median (min, max) instead of mean ± SD. Furthermore, please use Kruskal-Wallis test, instead of ANOVA.
6) Further, I suggest the authors present at least 1 Table with results if possible; although plots are interesting, Tables present the results with more accuracy for the reader.
7) In the discussion section (probably at the end) a part should be included dedicated to the drawbacks and weaknesses of the study.
8) The action of AST-120 in chronic kidney disease and especially the disease progression and cardiovascular complications (which influence prognosis) should be discussed more extensively. In this context, some additional studies about AST-120 regarding its action in chronic kidney disease should be included in the references.
Minor points
9) In the footnote/legend of the Figures please report the statistical tests used.
10) Verify that the legends of Figures 3E and 5D are correct and complete, respectively.
11) Correction of some grammatical and syntactical errors.
12) The terms “in vitro” and “in vivo” should be written in italics throughout the text and in the title.
Comments on the Quality of English LanguageModerate editing of English language required
Author Response
We deeply appreciate the reviewer’s positive comments of our manuscript. These comments have been addressed in a point-by-point manner as shown below.
(1) The scope section (end paragraph of introduction) somehow predefines the upcoming findings and the there from emerging discussion points. This is somehow confusing, especially expressions such as “we verified” and “the findings … illustrated” don’t belong in the Introduction.
Response to (1): Thanks for reviewer’s suggestion. We have corrected the descriptions in the scope section of this revised manuscript according to the suggestion of reviewer.
"This study was designed to examine whether IS induces ROS-related ferroptosis and its impact on iron metabolism in renal cells, and studied the role of ferroptosis in the IS-induced renal injury using a renal tubular cell model and an adenine-induced CKD mouse model. The research not only illustrated the mechanisms through which IS induced ferroptosis but provided a promising therapeutic option for CKD patients."
(2) In the Materials and Methods sub-section 2.3 the groups of animals are defined imprecisely. The addition of a sub-section entitled experimental design one for the cell cultures experiment and one for the animal experiments would add clarity for the reader.
Response to (2): Thanks for reviewer’s suggestion. We have re-written and added the experimental designs for cell cultures experiments and animal experiments in the Methods sub-sections 2.2 and 2.3 of this revised manuscript according to the suggestion of reviewer.
(3) The Results section includes not only the findings but also a lot of discussion and controversies about the findings. There are for example too many references that are part of the discussion of the findings. If this text and these references were added in the discussion it would make the discussion more extensive and would make the paper more clear cut.
Response to (3): Thanks for reviewer’s suggestion. We have deleted too many discussions in the results section and extend it to the discussion section of this revised manuscript according to the suggestion of reviewer.
(4) For example, the missing effect on NCOA4 both in RTC and in CKD mice is mentioned in the findings and discussed in a concise form there but is not even mentioned in the discussion section.
Response to (4): Thanks for reviewer’s suggestion. We have added the descriptions for the effects on NCOA4 both in RTC and in CKD mice in the Results section and the discussion about NCOA4 in the Discussion section of this revised manuscript according to the suggestion of reviewer.
(5) With a sample size of 6 mice per group, the authors should switch to non-parametric statistical analysis. Towards this end, please report median (min, max) instead of mean ± SD. Furthermore, please use Kruskal-Wallis test, instead of ANOVA.
Response to (5): Thanks for reviewer's suggestion. We have switched our statistical methods to non-parametric analysis- Kruskal-Wallis test in the animal experiments and have reported median (min, max) in the Bar Graphs in this revised manuscript according to the suggestion of reviewer.
(6) Further, I suggest the authors present at least 1 Table with results if possible; although plots are interesting, Tables present the results with more accuracy for the reader.
Response to (6): Thanks for reviewer’s suggestion. We have added one table (Table 1) to present serum biochemical indicators and total iron contents among different groups in CKD mice with or without AST120 treatment in this revised manuscript according to the suggestion of reviewer.
(7) In the discussion section (probably at the end) a part should be included dedicated to the drawbacks and weaknesses of the study.
Response to (7): Thanks for reviewer's suggestion. We have added the descriptions for the limitations of this study in the end of Discussion section of this revised manuscript according to the suggestion of reviewer.
(8) The action of AST-120 in chronic kidney disease and especially the disease progression and cardiovascular complications (which influence prognosis) should be discussed more extensively. In this context, some additional studies about AST-120 regarding its action in chronic kidney disease should be included in the references.
Response to (8): Thanks for reviewer’s helpful suggestion. We have added the information about AST-120 in the Discussion section of this revised manuscript according to the suggestion of reviewer.
" AST-120 is an oral activated carbon adsorbent that is widely used as a therapeutic to inhibit circulating uremic toxins in CKD patients [64], it provides renoprotection by preventing glomerular hypertrophy, fibrosis, proteinuria, and delay the initiation of dialysis in progressive CKD [35, 65]. In addition, AST-120 also improve bone health, preventing uremic osteoporosis by maintaining bone mass and reducing susceptibility to fractures [66, 67, 68]. Moreover, AST-120 address CKD-related inflammation by in-hibiting immune cell activation, oxidative stress, and inflammatory responses [69, 70]. It also protects endothelial function, enhances vasodilation and improves vascular health [71]. Notably, recent studies have shown that AST-120 exhibits potential cardi-oprotective effects by reducing aortic IS deposition, attenuating atherosclerosis, and protecting the heart and kidneys [72]. In the gastrointestinal environment, AST-120 restores mucosal integrity, corrects microbial imbalances, and alleviates renal decline [73, 74]. In summary, AST-120 has wide therapeutic effects. In the present study, we found that AST-120 can reduce the accumulation of iron in the kidneys of CKD mice and significantly improve kidney injury and fibrosis caused through senescence, ER stress, and ferroptosis, reconfirmed the therapeutic potential of AST-120 in the treat-ment of CKD. "
(9) In the footnote/legend of the Figures please report the statistical tests used.
Response to (9): Thanks for reviewer's suggestion. We have reported the statistical tests used in figure legends of this revised manuscript according to the suggestion of reviewer.
(10) Verify that the legends of Figures 3E and 5D are correct and complete, respectively.
Response to (10): Thanks for reviewer’s suggestion. We have corrected the legends of Figures 3E and 5D in this revised manuscript according to the suggestion of reviewer.
(11) Correction of some grammatical and syntactical errors.
Response to (11): Thanks for reviewer’s suggestion. We have revised the manuscript via the professional English editing by MDPI English pre-edit services (English Editing Invoice ID: english-71669).
(12) The terms “in vitro” and “in vivo” should be written in italics throughout the text and in the title.
Response to (12): Thanks for reviewer’s suggestion. We have corrected the terms “in vitro” and “in vivo” with italics throughout the text and in the title of this revised manuscript according to the suggestion of reviewer.
Reviewer 2 Report
Comments and Suggestions for Authors
The authors demonstrated that indoxyl sulfate (IS) aggravates renal injury due to alteration of iron metabolism and activation of reactive oxygen species (ROS) in cultured renal tubular cells and mouse chronic kidney disease (CKD) model. This manuscript is important and well written. However, there are some problems in this manuscript.
(1) The authors showed Figure 8 that ferroptosis induces senescence and ER stress in vitro experiments. However, there are no experiments to indicate the results. Therefore, the authors should show their results or revise Figure 8 adequately.
(2) Are there any drugs that can directly inhibit IS to better define its involvement in in vitro experiments? If such drugs are available, the authors should perform such experiments.
(3) Is it possible for the authors to prove that iron is involved in in vivo experiments using iron chelators? If such drugs are available, the authors should perform such experiments.
(4) Immunohistochemistry of Ferritin and HO-1 shows that all glomeruli, tubules, interstitium and vessels were stained strongly positive in the adenine-induced CKD model. However, it is questionable whether this is such a strong and non-site specific staining pattern. The authors should change to a more appropriate image. (5) The cell count is incorrectly indicated. The authors should revise them adequately.
Comments on the Quality of English LanguageBecause there are a few things wrong with the English description, it would be better to have someone who is a native English speaker correct it.
Author Response
We deeply appreciate the reviewer’s positive comments of our manuscript. These comments have been addressed in a point-by-point manner as shown below.
(1) The authors showed Figure 8 that ferroptosis induces senescence and ER stress in vitro experiments. However, there are no experiments to indicate the results. Therefore, the authors should show their results or revise Figure 8 adequately.
Response to (1): Thanks for reviewer’s suggestion. We have corrected Figure 8 to show that indoxyl sulfate (IS) triggers iron accumulation and elevates ROS levels, subsequently inducing ferroptosis, senescence and ER stress.
(2) Are there any drugs that can directly inhibit IS to better define its involvement in in vitro experiments? If such drugs are available, the authors should perform such experiments.
Response to (2): Thank you for reviewer’s suggestion regarding the use of drugs to inhibit indoxyl sulfate in our in vitro experiments. After searching the literature databases from PubMed® and Web of ScienceTM, there are no in vitro specific chelators or inhibitors that directly target and inhibit indoxyl sulfate (IS) itself.
(3) Is it possible for the authors to prove that iron is involved in in vivo experiments using iron chelators? If such drugs are available, the authors should perform such experiments.
Response to (3): We appreciate the reviewer's suggestion. We also agree that would be a valuable approach to investigate the in vivo effects of iron using iron chelators such as deferoxamine (DFO). However, it will require some time to obtain approval for the Animal Use Protocol from the Institutional Animal Care and Use Committee (IACUC). Therefore, we have no way to complete this in vivo experiment in the near future. However, please allow us to list this comment as one of the limitations of this study, which is described at the end of the Discussion section of this revised manuscript.
(4) Immunohistochemistry of Ferritin and HO-1 shows that all glomeruli, tubules, interstitium and vessels were stained strongly positive in the adenine-induced CKD model. However, it is questionable whether this is such a strong and non-site specific staining pattern. The authors should change to a more appropriate image. (5) The cell count is incorrectly indicated. The authors should revise them adequately.
Response to (4): Thank you for your valuable feedback regarding our immunohistochemistry analysis of Ferritin and HO-1 in the adenine-induced CKD model. We appreciate your input and understand your concern about the staining pattern observed in our study. We have carefully checked and revised the images that are more appropriate images to ensure the accuracy and specificity of our results. Although there was some nonspecific staining in the IHC results, we dedicated significant efforts to enlarging, refining, and adjusting the picture resolution to enhance the clarity of our data for analysis.
Response to (5): We appreciate the reviewer's suggestion. We utilized the ImageJ software with the IHC profiler plugin to quantify the IHC staining results, where staining intensity is typically scored on a scale ranging from 0 (negative) to 3 (high positive). We took this matter seriously and promptly checked and revised the data to ensure its accuracy. Moreover, according to the suggestion of reviewer 1, for animal experiments, we have switched our statistical methods to non-parametric analysis- Kruskal-Wallis test and have reported using median (min, max) in the Bar Graphs in this revised manuscript.
Round 2
Reviewer 2 Report
Comments and Suggestions for Authors
The authors adequately responded to my questions except for my fifth question. In addition, the authors' answer to my fifth question is not an answer to my question. However, fortunately, the error in the cell count I requested in my fifth question has been firmly corrected. Therefore, I have no special requests in this revised manuscript.